# Clinical Comparison Between Curative and Non-Curative Treatment for Hepatocellular Carcinoma with Hepatic Vein Invasion: A Nationwide Cohort Study

**DOI:** 10.3390/cancers17111794

**Published:** 2025-05-27

**Authors:** Sehyeon Yu, Hye-Sung Jo, Young-Dong Yu, Yoo-Jin Choi, Su-Min Jeon, Dong-Sik Kim

**Affiliations:** Division of Hepato-Biliary-Pancreatic Surgery and Liver Transplantation, Department of Surgery, Korea University College of Medicine, Seoul 02841, Republic of Korea; vict1216@naver.com (S.Y.); hust1351@naver.com (Y.-D.Y.); ujinny30@gmail.com (Y.-J.C.); suminjeon14@naver.com (S.-M.J.); kimds1@korea.ac.kr (D.-S.K.)

**Keywords:** hepatocellular carcinoma, hepatic vein invasion, curative-intent treatment

## Abstract

Hepatocellular carcinoma (HCC) with hepatic vein invasion (HVI) is typically treated with palliative approaches due to its advanced stage. However, this study shows that curative-intent treatments such as surgical resection or local ablation can significantly improve long-term survival in selected patients. Using data from a large Korean national registry, we compared outcomes between patients receiving curative versus non-curative treatment after adjusting for key clinical factors. Despite similar tumor burden, those in the curative group showed markedly better overall and cancer-specific survival. These findings suggest that curative approaches may offer a meaningful survival benefit for HCC even in the presence of HVI in patients with preserved liver function and good performance status.

## 1. Introduction

Hepatocellular carcinoma (HCC) is the most common primary liver cancer and one of the leading causes of cancer-related mortality worldwide [1,2,3]. Vascular invasion, observed in approximately 25–50% of HCC cases, is a well-established factor associated with poor prognosis [4]. The most commonly observed vascular invasion in HCC is portal vein invasion (PVI). Due to the biological feature for portal dissemination of HCC, PVI has been widely discussed. Hepatic vein invasion (HVI), although present in only about 5% of HCC patients [5], can cause direct dissemination of tumor cells into the systemic circulation [1,6]. Despite this concern, its prognostic relevance and therapeutic implications remain less well-defined than those of PVI.

The 8th edition of the American Joint Committee on Cancer (AJCC) staging system classifies HCC with major branch invasion of the hepatic vein as the T4 stage, whereas in previous editions, such major vascular involvement was categorized as T3b [7]. This reclassification reflected a poor prognosis associated with extensive vascular invasion. However, subsequent validation studies have not examined HVI independent of other T4 factors, such as direct invasion into adjacent organs [8,9,10]. Furthermore, few studies have specifically evaluated the prognostic impact or treatment strategies for HVI separately from PVI [11]. The Barcelona Clinic Liver Cancer (BCLC) staging system classifies HCC with PVI as advanced stage and recommends palliative treatment; however, specific guidelines for HCC with HVI remain insufficient [12]. A national practice guideline for the management of HCC states that in patients with well-preserved liver function, surgical resection may be selectively considered even in the presence of HVI. However, the level of evidence and strength of recommendation for this approach remains low [13].

Thus, clear guidelines regarding optimal treatment strategies for HCC with HVI are yet to be established. Considering the potential role of curative treatment for HCC with HVI, it is crucial to investigate the clinical impact of HVI and treatment-related outcomes using large-scale HCC registry data. Therefore, this study aimed to compare the clinical outcomes of curative and non-curative treatment in patients of HCC with HVI and to identify prognostic factors affecting long-term outcomes.

## 2. Materials and Methods

### 2.1. Data

We used the database of patients with HCC newly diagnosed in Korea between 2008 and 2019, provided by the Korean Primary Liver Cancer Registry (KPLCR). This study utilized a retrospective multicenter observational cohort comprising randomly extracted and registered HCC data from the Korean Central Cancer Registry (KCCR), based on the International Classification of Disease (ICD) code C22.0, collected since 2008. Information on patient survival status and censoring was obtained through the national death registry as of 31 December 2022.

### 2.2. Study Population and Matching

All patients diagnosed with HCC registered in the KPLCR from 54 hospitals between 2008 and 2019 were enrolled in this study (Figure 1). Among a total of 18,315 cases, patients who met the following criteria were excluded: (I) the presence of extrahepatic metastasis, (II) a Child-Turcotte-Pugh (CTP) score ≥ 7, (III) Eastern Cooperative Oncology Group (ECOG) performance status ≥ 2, and (IV) who underwent liver transplantation as 1st treatment for HCC. Of the remaining 12,211 patients, 350 had HVI at diagnosis, and 293 underwent treatment. After excluding 51 patients with insufficient medical records, a final analysis was conducted on 242 patients divided into two groups: a curative group (*n* = 42, 17.4%) comprising patients who were treated with surgical resection or local ablation therapy, and a non-curative group (*n* = 200, 82.6%), comprising patients who had received trans-arterial therapy, chemotherapy or radiation therapy.

However, patients of HCC with HVI receiving non-curative treatment are more likely to have a higher tumor burden and poor liver function, making them unsuitable candidates for curative-intent treatment. In addition, there was a substantial difference in patient numbers between the curative and non-curative groups. To minimize differences in baseline characteristics and sample size between the two groups, it was necessary to match critical factors related to liver function and morphologic tumor burden to ensure comparability. Therefore, we performed propensity score matching based on the albumin-bilirubin (ALBI) grade (grade 1: ≤ −2.60; grade 2: > −2.60 and ≤ −1.39; grade 3: > −1.39), the number and the maximum size of tumors for the curative and non-curative group. Finally, we compared and analyzed treatment outcomes of 42 (29.0%) and 103 (71.0%) patients in the curative and non-curative groups, respectively.

### 2.3. Diagnosis and Definition

All the patients included in the study were newly diagnosed with HCC. HCC was diagnosed if the histological and immunological findings were positive or if a tumor (or tumors) ≥ 1 cm in size was identified on multi-phase computed tomography (CT) or magnetic resonance imaging (MRI) with imaging features consistent with HCC: i.e., hyperenhancement in the arterial phase, and washout at the portal venous or delayed phase. HVI was assessed using imaging studies, and the provided dataset did not include information on the HVI grade.

Overall survival (OS) and cancer-specific survival (CSS) were analyzed in each group. OS was defined as the time from diagnosis to death or the last follow-up date, whereas CSS was defined as the time from diagnosis to death due to HCC, using the Korean Standard Classification of Diseases version 7 system or the last follow-up date.

### 2.4. Statistical Analysis

Categorical variables are presented as numbers with percentages compared using the chi-square test or Fisher’s exact test as appropriate. Continuous variables are presented as medians and interquartile ranges and compared using the Student’s *t*-test and Mann-Whitney test. Propensity score matching (PSM) was conducted using the ALBI score and the number and maximum size of tumors. After propensity score matching, the standard mean difference suggested an appropriate balance between the two groups. (Appendix A and Appendix A). Kaplan-Meier analysis and log-rank tests were used to calculate OS and CSS. Cox proportional hazards regression analysis was used to evaluate the prognostic factors for survival and was conducted in two stages: univariate and multivariate analyses. For the multivariate analysis, factors with a *p*-value of 0.150 or less in the univariate analysis were included, and factors with a *p*-value less than 0.050 were considered significant. Regarding time-dependent effects of risk factors on OS and CSS, the cumulative regression coefficients for each covariate were plotted with 95% confidence intervals to visualize the temporal contribution of each variable to the hazard function. Statistical analyses were performed using IBM Statistics for Windows (version 20.0; IBM Corp., Armonk, NY, USA) and R software (version 3.6.1; Vienna, Austria; “rms”, “survival” and “survminer” packages).

## 3. Results

### 3.1. Descriptive Analysis and Pre-Matched Cohort

The incidence of HCC with HVI was 5.4%, corresponding to 988 cases in the total cohort before applying the exclusion criteria and propensity score matching. The distribution of treatments administered to these patients is shown in Appendix A. Among them, only 58 (5.9%) patients received curative-intent treatment, whereas 541 (54.8%) received non-curative treatment. On the other hand, 360 (36.4%) patients who were diagnosed with HCC accompanied by HVI did not receive any treatment. Among them, 33 patients had CTP grade A and ECOG scores between 0 to 1 without extrahepatic metastasis. The median tumor number and size of those patients were 1 (1–5) and 12 (8.5–15.4) cm, respectively. The median alpha-fetoprotein (AFP) level was 201.5 (10.4–10,500.0) ng/mL (Appendix A).

Before PSM, the curative group comprised 42 (17.4%) patients, whereas the non-curative group comprised 200 (82.6%) patients. The number of tumors was significantly higher in the non-curative group compared to the curative group (1 [1–1] vs. 2 [1–5], *p* < 0.001). Additionally, the maximum tumor size was significantly larger in the non-curative group (5.1 [3.1–10.7] vs. 8.3 [5.3–12.0] cm, *p* = 0.010) and albumin levels were significantly higher in the curative group (4.3 [3.9–4.5] vs. 4.0 [3.6–4.2] g/dL, *p* < 0.001). These variables are related to tumor burden and underlying liver function and are considered potential factors that may significantly impact long-term survival. Therefore, we performed propensity score matching using tumor number, tumor size, and ALBI grade.

### 3.2. Baseline Characteristics of Matched Cohort

The baseline characteristics of the curative and non-curative groups after PSM are presented in Table 1. No statistically significant differences between the two groups in terms of age, sex, or major underlying conditions, including liver disease (*p* > 0.050) were observed. Serum albumin levels remained significantly higher in the curative group than in the non-curative group (4.3 [3.9–4.5] vs. 4.1 [3.8–4.3] g/dL, *p* = 0.035); however, there was no significant difference in ALBI grade between the two groups (*p* = 0.403). None of the patients had a history of hepatic encephalopathy, and there were no significant differences in laboratory tests between the two groups except for serum albumin levels. In addition, there were no cases of hepatic encephalopathy (HEP) in either group, and there were no significant differences between the two groups in portal hypertension parameters such as platelet count and ascites.

Regarding tumor burden-related factors, the number and size of tumors used for matching showed no differences between the curative and non-curative groups (1 [1–1] vs. 1 [1–2], *p* = 0.672 and 5.1 [3.1–10.7] vs. 7.3 [4.9–10.0] cm, *p* = 0.143, respectively) (Table 2). Additionally, there were no significant differences between the two groups in terms of tumor markers (alpha-fetoprotein and protein induced by vitamin K absence or antagonist-II), vascular or bile duct invasion as shown in Table 2 (*p* > 0.050).

### 3.3. Overall and Cancer-Specific Survival and Risk Factors

Survival rates according to treatment modality in the entire cohort before applying the exclusion criteria and propensity score matching are presented in Appendix A. The 5-year OS and CSS rates in no HVI group were 49% and 53%, respectively. In the curative-intent treatment group, the 5-year OS and CSS rates were 44% and 49%, respectively. This contrasts with each non-curative treatment group, in which the 5-year OS and CSS rates were 4–9% and 6–13%, respectively (*p* < 0.001).

After applying exclusion criteria and PSM, the median follow-up period for OS and CSS was 18 months. The 1-, 3-, and 5-year OS rates in the curative group were significantly higher than those in the non-curative group (83%, 64%, and 48% vs. 59%, 24% and 15%, respectively; *p* < 0.001) (Figure 2a). The 1-, 3-, and 5-year CSS rates in the curative group were also significantly higher than those in the non-curative group (83%, 64%, and 55% vs. 59%, 25%, and 17%, respectively; *p* < 0.001) (Figure 2b).

In the multivariate Cox regression analysis (Table 3), the non-curative treatment was identified as a strong risk factor for both OS and CSS (hazard ratio and 95% confidence interval, 2.43 [1.50–3.92], *p* < 0.001 and 2.45 [1.49–4.01], *p* < 0.001, respectively). ALBI grade 2 or higher was also found to be a significant risk factor for both OS and CSS (2.00 [1.31–3.14], *p* = 0.001 and 2.07 [1.33–3.22], *p* < 0.001, respectively). Additionally, AFP ≥ 400 ng/mL was also identified as a significant risk factor for both OS and CSS (1.70 [1.13–2.56], *p* = 0.011 and 1.70 [1.12–2.59], *p* = 0.013, respectively). On the other hand, the number of tumors, tumor size, and portal vein invasion were not significant risk factors for survival. The forest plot showed the results of the multivariate Cox regression analyses for OS and CSS (Figure 3).

Figure 4 illustrates the time-dependent effects of the risk factors identified from the multivariate Cox proportional hazards regression analysis for OS and CSS. The cumulative effects of selected covariates over time were evaluated using Aalen’s additive regression model. Solid lines represent the estimated cumulative effects and shaded areas indicate the corresponding confidence intervals. It is noteworthy that the initial treatment decision (curative vs. non-curative) for HCC with HVI exhibited an increasing cumulative impact on survival risk over time.

## 4. Discussion

Clinically evident HVI in HCC can cause extrahepatic spread through systemic circulation. Furthermore, with respect to higher-grade HVI, hepatic congestion due to outflow obstruction can lead to impaired liver function and promotion of intrahepatic metastases [1,14]. Therefore, HCC with HVI is considered an advanced stage, which causes hesitation in pursuing curative-intent treatment. Currently, curative-intent treatment for HCC with HVI is rarely performed in real clinical practice, and there has been a prevailing atmosphere of prioritizing the non-curative treatment, including transcatheter arterial chemoembolization (TACE), chemotherapy, or radiation therapy without a clear treatment guideline [15,16,17].

The incidence of HCC with HVI was low; only 5.9% of the HCC with HVI cases underwent curative treatment, while 54.8% of the patients received non-curative treatment in the entire cohort. Only a few studies have reported the efficacy of surgical resection or ablation therapy on HCC with vascular invasion, including portal vein, hepatic vein, and inferior vena cava [18,19,20,21]. One study compared the median OS for hepatectomy and TACE in cases of HCC with HVI and the median OS was 4.47 and 1.58 years, respectively [11]. Moreover, based on a nationwide Japanese study, the Japan Society of Hepatology recently (5th JSH-HCC Guidelines) recommends that surgical resection is prioritized if feasible in cases of vessel invasion [22]. In several studies using non-curative treatment such as trans-arterial therapy, chemotherapy, or radiation therapy, either alone or in combination, the median OS was reported to be between 7.9 and 17.4 months [23,24,25,26]. Although we could not evaluate the preoperative images of the patients, which made treatment decisions difficult, considering the relatively poor long-term survival rate of patients undergoing non-curative treatment for HCC with HVI raises concern regarding the potential role of curative-intent treatment in these patients.

In this study, the curative group showed better long-term survival than the non-curative group. The clearly different survival outcomes between the two groups highlight the need to reexamine the role of curative-intent treatment for HCC with HVI. In the entire cohort before applying the exclusion criteria and propensity score matching, the curative-intent treatment group demonstrated a slightly lower but comparable survival curve compared to patients without HVI (Appendix A). These findings suggest that curative-intent treatment could potentially mitigate the adverse prognostic impact of HVI, achieving survival outcomes comparable to patients without HVI. Furthermore, considering that the 5-year CSS rates of the curative group exceed 50% after PSM, curative-intent treatment could have a potential role in achieving better long-term outcomes in selected patients of HCC with HVI. As imaging techniques for HCC, surgical strategies, and perioperative management continue to progress, it is important to enhance our understanding of HCC with HVI and move away from a passive approach to treatment based on systematic research.

In multivariate Cox proportional hazards regression analysis, non-curative treatment, male sex, ALBI grade ≥ 2, and AFP ≥ 400 ng/mL were independent risk factors increasing HR for OS and CSS. Figure 4 illustrates the time-dependent HR for independent risk factors, demonstrating that the risk associated with the initial choice between curative and non-curative treatment continued to increase over time, similar to other well-known risk factors for HCC. This finding emphasizes the importance of initial treatment selection in patients of HCC with HVI who are eligible for curative-intent treatment.

In this study, a higher-grade ALBI was the strongest independent risk factor for long-term survival following non-curative treatment. The ALBI score provides a simplified yet robust assessment of liver function using only two parameters—serum albumin and total bilirubin—while being a continuous variable that allows for a more precise evaluation than the CTP score. It has been extensively validated in previous studies, with established grade cut-off values demonstrating a significant clinical impact on liver function [27,28,29,30]. Additionally, it does not include subjective factors such as ascites or hepatic encephalopathy, making it a more objective indicator for evaluating liver function. Our results underscore the clinical importance of the ALBI score in HCC, where—unlike other malignancies—both tumor factors and individual patient liver function serve as crucial prognostic indicators.

The strength of this study lies in utilizing large-scale HCC registry data to investigate the incidence, treatment trends, and clinical outcomes in patients of HCC with HVI. Establishing the criteria for applying curative-intent treatment to patients of HCC with HVI will require a more systematic research process. The results do not suggest that all patients of HCC with HVI should undergo curative-intent treatments. However, unless there are specific contraindications—such as poor liver function, complicated underlying conditions, old age, poor ECOG status, or insufficient future remnant liver volume—it is important to consider the potential for curative-intent treatment in patients of HCC with HVI.

This retrospective study has several limitations. First, the final cohort size was small due to the low incidence of HCC with HVI; however, the low incidence of HVI observed in large-scale HCC cohorts is a meaningful result in itself. Second, there were differences in both the number of cases and the baseline characteristics between the two groups; To minimize these differences, we performed PSM focusing on liver function and tumor burden, which were considered the most critical confounding factors. Including more variables in the PSM could have enhanced the reliability of the study results; however, this was limited by the relatively small sample size of the HVI group that received curative treatment. Nevertheless, most baseline characteristics were well-balanced between the groups following PSM. Moreover, one of the limitations of this study using a nationwide registry is the absence of detailed operative and pathological variables, including the grade of HVI. Nevertheless, because lower-grade HVI is often difficult to diagnose both radiologically and pathologically in clinical practice, a substantial proportion of patients in this study likely had high-grade HVI. In addition, as the data in this study were collected from patients treated between 2008 and 2019, a key limitation is that it does not reflect recent advances in systemic treatment strategies or their therapeutic outcomes [31]. Finally, due to the low incidence of HCC with HVI and retrospective study design, it is difficult to provide a detailed description of surgical indications for HCC with HVI. However, through PSM, we compared treatment outcomes between the two groups with similar liver function, tumor size, and tumor number, and found a statistically significant survival benefit in those receiving curative-intent treatment. These findings could serve as a foundation for more precise studies in the future.

## 5. Conclusions

This large-scale registry-based study demonstrated the significant potential of curative-intent treatment for HCC in patients accompanied by HVI with preserved liver function and performance status. Based on this study, well-designed prospective trials are warranted to establish precise selection criteria for identifying patients who are most likely to benefit from curative strategies and achieve improved survival outcomes.

## Figures and Tables

**Figure 1 cancers-17-01794-f001:**
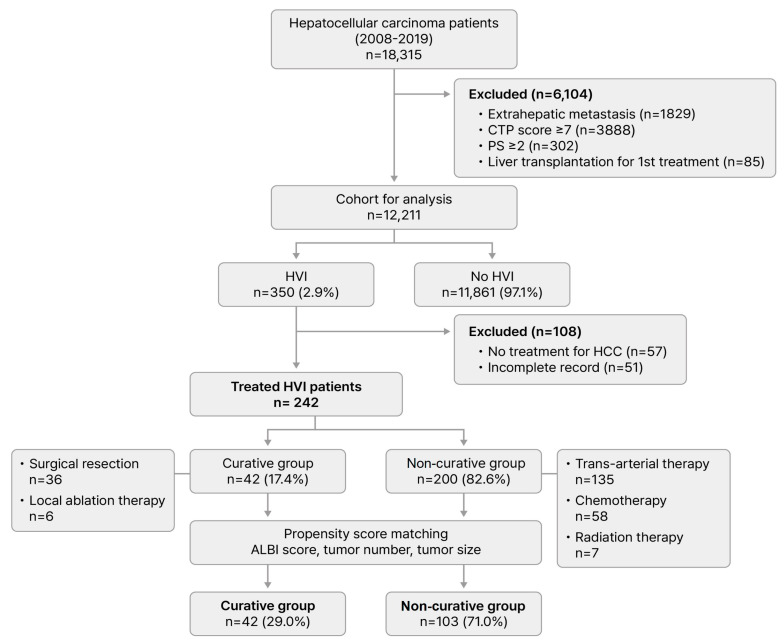
Study population. CTP, Child-Turcotte-Pugh; PS, performance status; HVI, hepatic vein invasion; HCC, hepatocellular carcinoma; ALBI, albumin-bilirubin.

**Figure 2 cancers-17-01794-f002:**
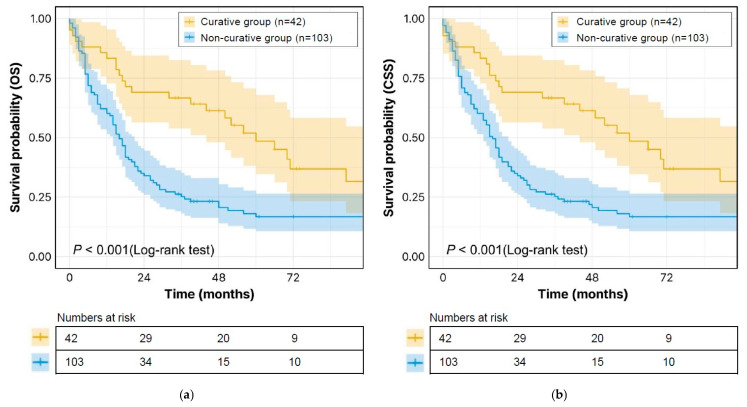
Survival curves for long-term outcomes in the curative group and the non-curative group. (**a**) OS; (**b**) CSS. Abbreviations: OS, overall survival; CSS, cancer-specific survival.

**Figure 3 cancers-17-01794-f003:**
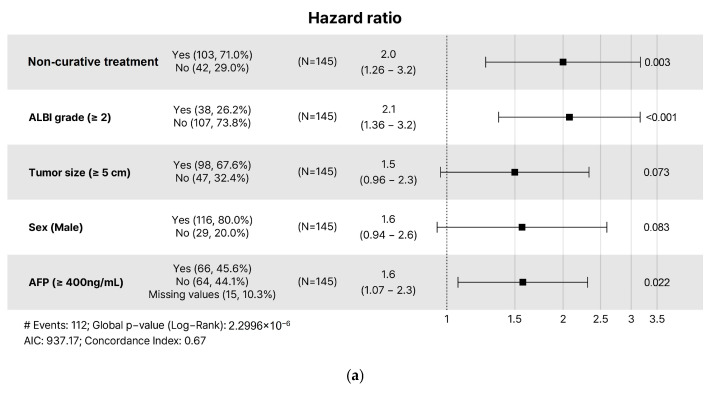
Forest plot (Cox-multivariate) for (**a**) OS and (**b**) CSS. Abbreviations: ALBI, albumin-bilirubin grade; AFP, alpha-fetoprotein; OS, overall survival; CSS, cancer-specific survival.

**Figure 4 cancers-17-01794-f004:**
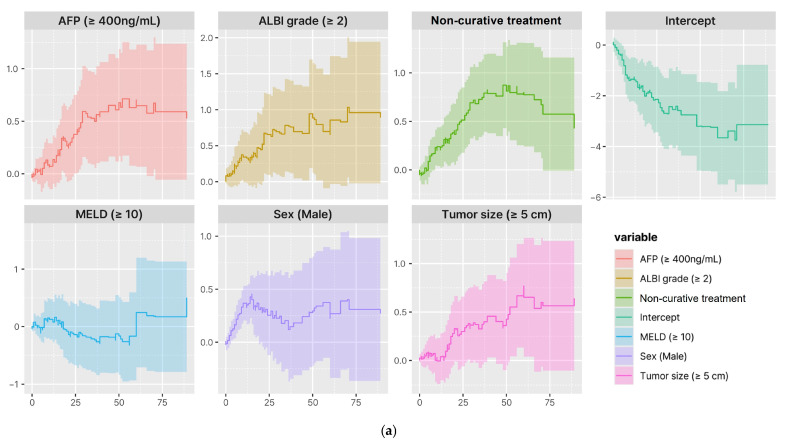
Time-varying effects of risk factors on OS and CSS. (**a**) Time-varying effects of covariates on OS; (**b**) Time-varying effects of covariates on CSS. The figure shows the cumulative effects of selected covariates over time based on Aalen’s additive regression model. Solid lines represent the estimated cumulative effects, and shaded areas indicate the confidence intervals. Abbreviations: AFP, alpha-fetoprotein; ALBI, albumin-bilirubin grade; MELD, a model for end-stage liver disease; OS, overall survival; CSS, cancer-specific survival.

**Table 1 cancers-17-01794-t001:** Baseline characteristics of curative and non-curative groups.

	Curative Group (*n* = 42)	Non-Curative Group (*n* = 103)	Total (*n* = 145)	*p*-Value
Age	63 (54–66)	60 (51–69)	61 (52–68)	0.827
Sex (female)	11 (26.2%)	18 (17.5%)	29 (20.0%)	0.257
BMI (kg/m^2^)	23.8 (21.7–26.0)	23.5 (20.9–25.5)	23.7 (21.4–25.7)	0.553
Smoking	24 (57.1%)	51 (50.5%)	75 (52.4%)	0.468
PST (grade 1)	8 (26.7%)	23 (28.7%)	31 (28.2%)	0.829
Hypertension	16 (38.1%)	43 (41.7%)	59 (40.7%)	0.685
Diabetes mellitus	9 (22.0%)	27 (26.7%)	36 (25.4%)	0.553
**Underlying liver disease (multiple)**				
HBV	28 (66.7%)	64 (64.6%)	92 (65.2%)	0.818
HCV	4 (10.5%)	18 (19.6%)	22 (16.9%)	0.211
Alcoholic liver disease	14 (33.3%)	37 (35.9%)	51 (35.2%)	0.767
**Underlying liver function**				
Ascites ‡	3 (7.1%)	12 (11.7%)	15 (10.3%)	0.555
Hepatic encephalopathy	0 (0%)	0 (0%)	0 (0%)	—
MELD score †	7 (7–9)	8 (7–9)	8 (7–9)	0.200
ALBI grade †				0.403
1; ≤−2.60	33 (78.6%)	74 (71.8%)	107 (73.8%)	
2; <−2.60 and ≤−1.39	9 (21.4%)	29 (28.2%)	38 (26.2%)	
3; >−1.39	–	–	–	
**Laboratory findings**				
Albumin (g/dL)	4.3 (3.9–4.5)	4.1 (3.8–4.3)	4.1 (3.8–4.4)	0.035
Total bilirubin (mg/dL)	0.80 (0.50–1.11)	0.74 (0.52–1.10)	0.80 (0.52–1.10)	0.867
PT-INR	1.04 (1.00–1.12)	1.08 (1.01–1.13)	1.07 (1.01–1.13)	0.223
Platelet (×10^3^)	177 (141–218)	171 (124–222)	173 (128–219)	0.793
Cr (mg/dL) †	0.80 (0.71–0.95)	0.87 (0.72–1.03)	0.84 (0.72–1.00)	0.171

Values are presented as median (IQR) for continuous data and *n* (%) for categorical data. † Mann–Whitney, ‡ Fisher’s exact. Abbreviations: BMI, body mass index; PST, performance status test; HBV, hepatitis B virus; HCV, hepatitis C virus; CTP, Child-Turcotte-Pugh; MELD, model for end-stage liver disease; ALBI, albumin-bilirubin; PT-INR, prothrombin time-international normalized ratio; Cr, creatinine.

**Table 2 cancers-17-01794-t002:** Characteristics of HCC between curative and non-curative groups.

	Curative Group (*n* = 42)	Non-Curative Group (*n* = 103)	Total (*n* = 145)	*p*-Value
Tumor number	1 (1–1)	1 (1–2)	1 (1–1)	0.672
Tumor size (cm, maxinum)	5.1 (3.1–10.7)	7.3 (4.9–10.0)	6.8 (3.8–10.6)	0.143
Portal vein invasion	18 (42.9%)	55 (53.4%)	73 (50.3%)	0.250
Hepatic artery invasion ‡	1 (2.4%)	1 (1.0%)	2 91.4%)	0.497
Bile duct invasion ‡	0 (0%)	5 (4.9%)	5 (3.4%)	0.322
AFP (ng/mL) †	183.4 (15.6–2528.0)	496.0 (31.8–6501.3)	417.2 (29.0–4628.8)	0.153
PIVKA-II (mAU/mL)	290.0 (57.0–2324.0)	1709.5 (180.5–10,215.5)	861.0 (127.0–7289.0)	0.651

Values are presented as median (IQR) for continuous data and *n* (%) for categorical data. † Mann–Whitney, ‡ Fisher’s exact. Abbreviations: HCC, hepatocellular carcinoma; AFP, alpha-fetoprotein; PIVKA-II, protein induced by vitamin K absence or antagonist-II.

**Table 3 cancers-17-01794-t003:** Univariate and multivariate analysis for overall and cancer-specific survival in the matched cohort.

	Overall Survival				Cancer-Specific Survival			
	Univariate Analysis		Multivariate Analysis		Univariate Analysis		Multivariate Analysis	
	HR (95% CI)	*p*-Value	HR (95% CI)	*p*-Value	HR (95% CI)	*p*-Value	HR (95% CI)	*p*-Value
Non-curative treatment	2.19 (1.39–3.44)	0.001	2.43 (1.50–3.92)	<0.001	2.36 (1.47–3.81)	<0.001	2.45 (1.49–4.01)	<0.001
Age (≥65 years)	0.99 (0.68–1.46)	0.977			0.89 (0.60–1.33)	0.574		
Sex (male)	1.69 (1.03–2.77)	0.039	1.91 (1.10–3.33)	0.022	1.68 (1.01–2.80)	0.045	2.04 (1.17–3.58)	0.012
BMI (<18.5 kg/m^2^)	1.94 (0.78–4.77)	0.152			1.91 (0.77–4.71)	0.161		
Smoking	1.07 (0.73–1.56)	0.732			1.09 (0.74–1.61)	0.648		
PST (≥1)	1.13 (0.71–1.82)	0.603			1.17 (0.73–1.88)	0.523		
HBV	0.84 (0.57–1.24)	0.385			1.00 (0.67–1.50)	0.999		
HCV	1.21 (0.71–2.07)	0.483			1.27 (0.74–2.18)	0.383		
Alcoholic liver disease	0.92 (0.62–1.37)	0.678			0.90 (0.60–1.36)	0.625		
Ascites (positive)	1.25 (0.68–2.27)	0.473			1.33 (0.73–2.42)	0.359		
ALBI grade (≥2)	1.92 (1.28–2.88)	0.002	2.00 (1.31–3.14)	0.001	1.91 (1.26–2.89)	0.002	2.07 (1.33–3.22)	<0.001
MELD score (≥10)	1.48 (0.93–2.36)	0.095			1.33 (0.82–2.18)	0.252		
Multiple tumors (≥2)	0.97 (0.62–1.50)	0.880			1.03 (0.66–1.61)	0.892		
Tumor size (≥5 cm)	1.88 (1.23–2.90)	0.004	1.39 (0.87–2.23)	0.168	1.91 (1.22–2.96)	0.004	1.46 (0.90–2.36)	0.124
Portal vein invasion	1.21 (0.83–1.75)	0.327			1.30 (0.88–1.91)	0.183		
Hepatic artery invasion	0.60 (0.08–4.28)	0.607			0.66 (0.09–4.71)	0.675		
Bile duct invasion	1.05 (0.39–2.85)	0.929			1.08 (0.40–2.94)	0.880		
AFP (≥400 ng/mL)	1.35 (0.91–2.00)	0.131	1.70 (1.13–2.56)	0.011	1.44 (0.97–2.16)	0.073	1.70 (1.12–2.59)	0.013
PIVKA-II (≥500 mAU/mL)	1.35 (0.85–2.13)	0.199			1.43 (0.89–2.28)	0.136		

Abbreviations: BMI, body mass index; PST, performance status test; HBV, hepatitis B virus; HCV, hepatitis C virus; ALBI, albumin-bilirubin; MELD, model for end-stage liver disease; AFP, alpha-fetoprotein; PIVKA-II, protein induced by vitamin K absence or antagonist-II; HR, hazard ratio, CI, confidence interval.

## Data Availability

The data is unavailable due to privacy and ethical restrictions.

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
