# Peer review of "Clinical Comparison Between Curative and Non-Curative Treatment for Hepatocellular Carcinoma with Hepatic Vein Invasion: A Nationwide Cohort Study"

_cancers, 2025, doi:10.3390/cancers17111794_

Round 1

Reviewer 1 Report

Comments and Suggestions for Authors

THe manuscript is interesting and well written. THe authors should detail better which treatments were given, particularly concerning the systemic treatments.

The authors should comment on the potential available choices in this setting, in this regard cite the recent SRMA (PMID: 31877664)

Please add the number at risk to the KM curves

I have some concerns about the choice of the variables used to build the propensity score matching, and there is also some collinearity among the variables used

Author Response

Dear Reviewer 1,

We would like to express our sincere gratitude for your thoughtful review of our manuscript. We truly appreciate your insightful suggestions and constructive comments.

We have made every effort to respond to your comments with great care and attention. Please kindly refer to the uploaded file below for our detailed responses.

Once again, we deeply thank you for your valuable time and support.

Respectfully,
The authors

Reviewer 2 Report

Comments and Suggestions for Authors

Thank you for the opportunity to review this interesting nationwide registry analysis comparing curative‑intent and palliative therapies for hepatocellular carcinoma (HCC) with hepatic vein invasion (HVI). The topic is clinically important, and the dataset is large. Below are my detailed comments.

  • The “palliative group” pools TACE, systemic therapy and radiation—modalities with widely disparate prognoses. Please report subgroup outcomes (e.g. TACE alone vs systemic therapy) and consider adjusting for treatment type in multivariable models or providing landmark analyses.
  • Absence of HVI grade and operative/pathological data limits interpretability. At minimum, acknowledge this as a key limitation and discuss whether low‑grade HVI might have been over‑represented in the curative group.
  • Specify the source of mortality data (national death registry, hospital records, linkage methodology) and the censoring date.
  • 360 patients with HVI received no therapy; 33 of them appeared potentially resectable (CTP A, ECOG 0‑1, no metastasis). Explain why they were excluded and discuss how their omission might bias results.
  • Add explicit statements on retrospective design, unmeasured confounders (e.g. platelet count, portal hypertension parameters), lack of liver‑transplant information post‑index treatment, and evolution of systemic therapy since 2019 (e.g. immunotherapy).

Author Response

Dear Reviewer 2,

We would like to express our sincere gratitude for your thoughtful review of our manuscript. We truly appreciate your insightful suggestions and constructive comments.

We have made every effort to respond to your comments with great care and attention. Please kindly refer to the uploaded file below for our detailed responses.

Once again, we deeply thank you for your valuable time and support.

Respectfully,
The authors

Reviewer 3 Report

Comments and Suggestions for Authors

Comments on hepatocellular carcinoma with hepatic vein invasion

In the staging system of BCLC classification, if the hepatocellular carcinomas (HCC) are found to have vascular (including portal vein, hepatic vein and hepatic artery) invasion, they are called grade C. It is almost a common sense that when facing cases of HCC of grade C, surgical intervention will not be applicable, the choice of management should be other locoregional methods such as TAE and radiotherapy or systemic treatment including chemotherapy and immuno-oncological therapy.

In this manuscript, the situation of patients of HCC with hepatic vein invasion (HVI) undergoing surgical treatment subverts the above-mentioned consensus. This result will be interesting to the readers.

However, in addition to having higher serum albumin level, inspecting the table 2, there seemed to be a tendency (although not statistically significant ) that curative group had smaller tumor size and lower AFP value, compared with those of palliative group.

For these extraordinary 42 patients, before this article could be considered for acceptance, I think that the authors need to provide more detailed information, beside level of AFP and tumor size, at least the tumor location, the status of concomitant portal vein invasion (survival between with and without) should be descried more clearly. The authors should review the data of these patients and try to elicit more precise or defined guidelines to tell clinician what condition the patients could be arranged for surgical intervention. The conclusion only said that ‘significant potential… in selected patients ’ is not enough.

The term ‘palliative’ in the topic and the content is not correct. Locoregional or systemic treatment are all aggressive modalities, not palliation. 

Author Response

Dear Reviewer 3,

We would like to express our sincere gratitude for your thoughtful review of our manuscript. We truly appreciate your insightful suggestions and constructive comments.

We have made every effort to respond to your comments with great care and attention. Please kindly refer to the uploaded file below for our detailed responses.

Once again, we deeply thank you for your valuable time and support.

Respectfully,
The authors

Round 2

Reviewer 1 Report

Comments and Suggestions for Authors

THe manuscript is OK now. Thank you!

Reviewer 3 Report

Comments and Suggestions for Authors

I accepted the explanations and revisions from the authors.